# The Role of MicroRNAs in Epidermal Barrier

**DOI:** 10.3390/ijms21165781

**Published:** 2020-08-12

**Authors:** Ai-Young Lee

**Affiliations:** Department of Dermatology, College of Medicine, Dongguk University Ilsan Hospital, 814 Siksa-dong, Ilsandong-gu, Goyang-si, Gyeonggi-do 410-773, Korea; lay5604@naver.com; Tel.: +82-319617250; Fax: +82-319617695

**Keywords:** miRNAs, skin barrier integrity, epidermal cell differentiation and proliferation, cell–cell adhesion, skin lipids

## Abstract

MicroRNAs (miRNAs), which mostly cause target gene silencing via transcriptional repression and degradation of target mRNAs, regulate a plethora of cellular activities, such as cell growth, differentiation, development, and apoptosis. In the case of skin keratinocytes, the role of miRNA in epidermal barrier integrity has been identified. Based on the impact of key genetic and environmental factors on the integrity and maintenance of skin barrier, the association of miRNAs within epidermal cell differentiation and proliferation, cell–cell adhesion, and skin lipids is reviewed. The critical role of miRNAs in the epidermal barrier extends the use of miRNAs for control of relevant skin diseases such as atopic dermatitis, ichthyoses, and psoriasis via miRNA-based technologies. Most of the relevant miRNAs have been associated with keratinocyte differentiation and proliferation. Few studies have investigated the association of miRNAs with structural proteins of corneocytes and cornified envelopes, cell–cell adhesion, and skin lipids. Further studies investigating the association between regulatory and structural components of epidermal barrier and miRNAs are needed to elucidate the role of miRNAs in epidermal barrier integrity and their clinical implications.

## 1. Introduction

MicroRNAs (miRNAs) are short, non-coding endogenous single-stranded RNA molecules, each composed of 19–25 nucleotides that mostly cause target gene silencing via transcriptional repression and/or degradation of target mRNAs. MiRNAs have been reported to target over one-third of human genes [1,2]. Thus, miRNAs act as essential regulators of a plethora of cellular activities, such as cell growth, differentiation, development, and apoptosis [3]. The key role of miRNAs in controlling mammalian skin development has been identified in a mouse study via the epidermal-specific deletion of key enzymes in the miRNA biogenesis pathways, including Dicer and Dgcr8 (DGCR8 microprocessor complex subunit; cofactor of drosha ribonuclease III) [4,5]. These mice manifest phenotypes of dehydrated skin, apoptotic hair follicles, and neonatal lethality. An increasing number of studies have supported that miRNAs are involved in morphogenesis and homeostasis of the skin and its appendages [6]. The miRNAs act as essential regulators of differentiation, proliferation, and survival at the cellular level of skin cells including keratinocytes [7,8] suggesting the role of miRNA in epidermal barrier integrity. 

The skin is primarily the first barrier against external insults, protecting against mechanical insults, microorganisms, chemicals, and allergens. It is a complex barrier system for defense. The stratum corneum, the outermost layer of skin, which plays a main role in the formation of skin barrier, consists of several layers of corneocytes with cornified envelopes, corneodesmosomes, and intercellular lipid lamellae. Tight junctions in the granular cell layer also contribute to the mechanical barrier by regulating the selective permeability of the paracellular pathway for calcium reabsorption. Tight junctions are composed of transmembrane proteins including claudin-1 (CLDN1). However, the association between claudins with miRNAs has yet to be investigated in the skin.

Ichthyoses represent skin diseases associated with primary barrier dysfunction under pathologic conditions [9]. However, limited data demonstrate the role of miRNAs in ichthyoses. Atopic dermatitis is another disease representing abnormal skin barrier function. The miRNAs are also associated with type 2 T helper (Th2) immune responses, one of the main mechanisms involved in the pathogenesis of atopic dermatitis, whereas few data are related to skin barrier function [10,11,12]. More studies have investigated skin disorders characterized by aberrant keratinocyte proliferation, which is also important in the formation and maintenance of skin barrier function, such as psoriasis, cutaneous wound healing and skin cancers.

This review covers the role of miRNAs in the formation and maintenance of skin barrier and its integrity in normal and diseased conditions, under the impact of genetic and environmental factors in epidermal cell differentiation and proliferation, cell–cell adhesion, and skin lipid synthesis.

## 2. Association of miRNAs with Keratinocyte Differentiation and Proliferation

To explore the relationship between miRNAs and keratinocyte differentiation and proliferation, the review highlights the role of regulatory factors, keratinocyte proliferation, and structural proteins in corneocytes and cornified envelopes.

### 2.1. Factors Regulating Keratinocyte Differentiation and Proliferation

The formation of calcium gradients across the epidermis coincides with developmental milestones of keratinocyte proliferation and differentiation [13]. Keratinocyte differentiation occurs whenever cells irreversibly exit the cell cycle after mitosis in the basal layer in contrast to keratinocyte proliferation. Transcriptional factors p63 and Notch are also involved in the regulation of epidermal keratinocyte differentiation and proliferation [14,15,16]. Notch activity removes keratinocytes from the cell cycle, whereas the absence of Notch activity induces hyperproliferation via persistent Wnt/β-catenin signaling [17]. The association between miRNAs and regulatory factors associated with keratinocyte differentiation and proliferation is discussed (Table 1).

#### 2.1.1. Epidermal Calcium Gradients

Mammalian epidermis shows a characteristic calcium gradient. Calcium levels are low in basal and spinous layers, but increase gradually towards the granular layers. Calcium levels decline again in the stratum corneum. Epidermal calcium gradients are formed mainly by Ca^2+^ influx from extracellular sources and Ca^2+^ release from endoplasmic reticulum stores via calcium-sensing receptor (CaR) and epidermal calcium channels including store-operated calcium entry (SOCE) channels, respectively. Stromal interaction molecule1 (STIM1), an essential component of SOCE in human keratinocytes, is a calcium sensor of endoplasmic reticulum.

Multiple miRNAs have been associated with the differentiation of human keratinocytes by extracellular calcium [28,37]. Several studies involved miR-203, which is the first and the most upregulated miRNA implicated in epidermal differentiation. MiR-203, which is expressed primarily in suprabasal keratinocytes [38], regulates calcium-induced keratinocyte differentiation by activation of the protein kinase C (PKC) and activator protein 1 (AP-1) pathway [37] via two transcription factors, snail family transcriptional repressor 2 (SNAI2) and ΔNp63, as targets [18]. The inhibition of miR-203 in p63 expression has also been implicated in the regulation of keratinocyte proliferation and differentiation by galectin-7, which upregulates the c-Jun N-terminal kinase (JNK) [19]. Oleic acid, an unsaturated free fatty acid constituent of sebum, has been shown to upregulate miR-203 expression to induce keratinocyte differentiation along with involucrin expression by targeting p63 [20]. MiR-203 targets the 3’-UTR of p63 in mouse skin and promotes cell cycle exit, although co-suppression of other miR-203 targets, S-phase kinase associated protein 2 (Skp2) and Musashi RNA-binding protein 2 (Msi2), is required for adequate induction of cell cycle exit [21]. The levels of miR-574 and miR-720 are increased in keratinocytes cultured at high calcium concentrations with increased differentiation markers including involucrin. p63 is also a direct target of these miRNAs. These miRNAs represent direct targets of iASPP, an inhibitory member of the apoptosis-stimulating protein of p53 (ASPP) family in keratinocytes. Silencing of iASPP in keratinocytes induces their differentiation via upregulation of miR-574 and miR-720, thereby downregulating p63 [22]. The actin cytoskeleton is remodeled during keratinocyte differentiation. Overexpression of miR-24 in keratinocytes, which control actin filament formation, triggers keratinocyte differentiation by targeting cytoskeletal modulators including p21-activated kinase 4 (PAK4) [23]. MiR-23b is associated with keratinocyte differentiation in vivo and in vitro, and is a valid differentiation marker of human keratinocytes [28], although the mechanism of miR-23b has been investigated in tumorigenesis rather than keratinocyte differentiation. The Sma- and Mad-related (SMAD) family of proteins act as transcription factors in the transforming growth factor-β (TGF-β) superfamily signaling pathway. The role of TGF-β/SMAD signaling in the regulation of human epidermal differentiation has been identified in three-dimensional human skin cultures [39]. The transforming growth factor-β-induced factor homeobox 1 (TGIF1), which interferes with TGF-β signaling, has been identified as a direct target of miR-23b-3p. Repression of TGIF1 by miR-23b-3p activates TGF-β signaling via phosphorylation of SMAD family protein, leading to keratinocyte differentiation [24]. NK3 homeobox 1 (NKX3.1) has been identified in epidermal keratinocytes. Keratinocytes exposed to high calcium can upregulate miR-378b to stimulate keratinocyte differentiation by targeting NKX3.1 [25]. Defective barrier function is related to aged epidermis. Investigation of the role of miRNAs in age-related keratinocyte differentiation and barrier function revealed that the upregulation of miR-30a in human keratinocytes treated with calcium [28] results in impaired keratinocyte differentiation and barrier defects mediated via lysyl oxidase (LOX), isocitrate dehydrogenase 1 (IDH1), and apoptosis and caspase activation inhibitor (AVEN) [26].

CaR and epidermal calcium channels require Ca^2+^ influx and release [40]. However, the expression of these receptors and channels has rarely been identified in connection with miRNAs in keratinocytes. Extracellular Ca^2+^ increases miR-184 expression in primary epidermal keratinocytes in a SOCE-dependent manner. The upregulated miR-184 facilitates keratinocyte differentiation with increased involucrin expression via upregulation of cyclin E and p21 cyclin-dependent kinase inhibitors [27].

#### 2.1.2. p63

The transcription factor p63, encoded by the *TP63* gene, is an important player in epidermal keratinocyte proliferation and differentiation. The two main isoforms of p63 include isoforms with the transactivation (TA-) domain and isoforms truncated at the NH2-terminus lacking the TA domain (ΔN-). ΔNp63α is the predominant isoform present in adult human epidermis and its expression is associated with skin proliferation. 

The gene expression is affected by p63 via multiple mechanisms including regulation of non-coding RNAs including miRNAs [41]. As for the association with miRNAs, p63 acts as a direct target of miRNAs such as miR-203 [18,19,20,21], miR-574, and miR-720 [22] in calcium-induced keratinocyte differentiation. Hailey-Hailey disease, an autosomal dominant disorder characterized by suprabasal acantholysis, is a cornification disorder associated with mutations in the ATPase secretory pathway Ca^2+^ transporting 1 (ATP2C1) gene encoding intracellular calcium pumps resulting in abnormal cytosolic Ca^+2^ levels [42]. The role of miR-125b in the pathogenesis of Hailey-Hailey disease highlights p63 as a target of miRNAs. MiR-125b, which is increased via an oxidative stress-dependent mechanism, is involved in keratinocyte differentiation and proliferation via suppression of both p63 and Notch1 expression [30].

However, p63 targets miRNAs to regulate the proliferation and differentiation of keratinocytes by directly binding to both miR-34a and miR-34c regulatory regions and repressing their activity, resulting in keratinocyte proliferation [43]. Upregulation of miR-34a stimulates keratinocyte differentiation via the downregulation of SIRT6, a sirtuin family member, as a direct target [34,35]. Expression of epidermal differentiation complex protein is regulated by a pool of transcription factors, including Kruppel-like factor 4 (KLF4) [44]. Upregulation of miR-34a, which has also been associated with physiological skin aging, inhibits the expression of KLF4 by inducing a senescent phenotype in primary keratinocytes [36].

ΔNp63 also enhances the activity of miRNAs and is involved in early steps of keratinocyte differentiation. Intron 4 of human TP63 contains the gene for miR-944, which is highly expressed in keratinocytes. The promoter of an intronic miRNA, miR-944, is activated by binding of ΔNp63. As a target of ΔNp63, miR-944 upregulates the expression of keratin 1 (K1) and K10, but not filaggrin, involucrin, loricrin and transglutaminases, regulating the early steps of epidermal differentiation by inhibiting ERK signaling and increasing p53 expression [31]. Similarly, p63 silencing downregulates multiple miRNAs, particularly members of the miR-17 family such as miR-17, miR-20b and miR-106a in keratinocytes, which play a role in the onset of keratinocyte differentiation with K1 and K10 expression via upregulation of p21, retinoblastoma 1 (RB1 1), and JNK2 targets [32].

#### 2.1.3. Notch Signaling

In addition to primary epidermal keratinocytes under high calcium concentrations, the regulatory role of miR-184 in balance between keratinocyte differentiation and proliferation has been identified using loss-of-function and gain-of-function animal models. Upregulation of miR-184 promotes keratinocyte differentiation by enhancing the Notch pathway and targeting K15 and factor-inhibiting hypoxia-inducible factor 1 (FIH1) [33]. The discovery of abnormal differentiation and hyperproliferation of epidermal keratinocytes in psoriasis pathogenesis has greatly facilitated the correlation between miRNAs and Notch signaling [45]. Therefore, the relevant findings are reviewed in connection with the keratinocyte proliferation. 

### 2.2. Keratinocyte Proliferation

A balance between keratinocyte proliferation and differentiation is important in creating and maintaining intact skin barrier function. Aberrant keratinocyte proliferation in skin disorders, such as psoriasis and wound healing, can have a negative effect on the barrier function. 

Substantial data associated with miRNAs, either upregulated or downregulated, and keratinocyte proliferation have been reported in psoriasis. The downregulated miRNAs include miR-125b, miR-181b-5p, miR-520a, miR-194, miR-217, miR-138, miR-320b, miR-150, miR-145-5p, miR-20a-3p, miR-876-5p, miR-99a, miR-187, miR-548a-3p, miR-330, and miR-146a. The upregulated miRNAs identified include miR-21, miR-31, miR-744-3p, miR-17-92, miR-130a, miR-122-5p, miR-223. The downregulation of miR-96-5p and upregulation of miR-21, miR-31, miR-17-3p, and miR-126 in wound healing are associated with keratinocyte proliferation. The downregulation of miR-181a and miR-146a and upregulation of miR-21 and miR-31 in cutaneous squamous cell carcinoma (cSCC) play a role in keratinocyte proliferation. The expression of miR-99b and miR-203 is downregulated in keratinocyte proliferation of condyloma acuminatum and epidermodysplasia verruciformis (Table 2).

#### 2.2.1. Psoriasis

Downregulation of miR-125b stimulates keratinocyte proliferation and aberrant differentiation by targeting the fibroblast growth factor receptor 2 (FGFR2) [46]. AKT serine/threonine kinase 3 (protein kinase B gamma, AKT3) has also been identified as a direct target of miR-125b. Downregulation of miR-181b-5p also occurs in keratinocyte proliferation by targeting AKT3 [47]. AKT, known as protein kinase B, is also identified as a direct target of miR-520a [49]. The decreased expression of miR-320b promotes keratinocyte proliferation by enhancing AKT3 target [50]. Although toll-like receptor 4 (TRL4) mediates the immune response in psoriasis, the regulatory role of TLR4 in keratinocyte proliferation in psoriasis has been identified as a direct target of downregulated miR-181b [48]. Grainyhead-like 2 (GRHL2) regulates epithelial cell proliferation and differentiation as an epithelial-specific transcription factor. Downregulated miR-194 contributes to proliferation via GRHL2 target [51]. GRHL2 has also been identified as a direct target of miR-217 [52]. Telomere shortening prevents aberrant cell proliferation. Telomerase reverse transcriptase, known as TERT, which is highly activated in many cancers, adds telomere repeats to the chromosome end. GRHL2 is necessary for the expression of human telomerase reverse transcriptase (hTERT), the catalytic subunit of human telomerase [84]. Reduced levels of miR-138 stimulate keratinocyte proliferation by targeting hTERT [53]. Based on local hypoxia and vascular abnormal growth characteristics of psoriasis with abnormal proliferation of keratinocytes, the role of downregulated miR-150 in keratinocyte proliferation has been identified by targeting hypoxia-inducible factor1 subunit α (HIF-1α) and vascular endothelial growth factor A (VEGFA) [54]. Downregulated miR-145-5p stimulates keratinocyte proliferation by targeting mixed-lineage kinase 3 (MLK3) [55], which regulates the JNK and p38 signaling pathways. Loss of miR-20a-3p plays a role in psoriasis by targeting scm-like with four mbt domains 1 (SFMBT1) [56] underlying multiple cellular phenomena, including cell proliferation. Downregulated miR-876-5p is involved in keratinocyte proliferation by repressing a direct target angiopoietin-1 (ANG-1) [57]. Downregulation of miR-99a enhances keratinocyte proliferation by targeting FZD5 (Frizzled-5)/FZD8 via downstream factors β-catenin and cyclinD1 [58]. The role of miR-187 downregulation in keratinocyte proliferation has been demonstrated by targeting CD276 [59]. IL-22 plays a critical role in the pathogenesis of psoriasis. Downregulated miR-548a-3p targets protein phosphatase 3 regulatory subunit B, alpha (PPP3R1) in IL-22 mediated keratinocyte proliferation [60]. Downregulated miR-330 also contributes to IL-22-mediated keratinocyte proliferation by activating catenin beta 1 (CTNNB1) [61]. Downregulation of miR-146a in psoriasis promotes keratinocyte proliferation via the activation of a direct target, epidermal growth factor receptor (EGFR) [62]. 

However, the upregulation of miR-122-5p induces IL-22-mediated keratinocyte proliferation by targeting sprouty homolog 2 (SPRY2) [80]. Upregulation of miR-223 also promotes proliferation of IL-22-stimulated keratinocytes by targeting phosphatase and tensin homolog (PTEN) [81]. Other upregulated miRNAs in psoriasis, such as miR-21, miR-31, miR-744-3p, miR-17-92 and miR-130a, have been involved in keratinocyte proliferation. Upregulated miR-21, which is considered as an oncogene, stimulates keratinocyte proliferation via downregulation of caspase-8 [67,68]. Although miR-31 upregulates the expression of an endogenous negative regulator of factor-inhibiting hypoxia-inducible factor 1 (hypoxia-inducible factor 1, subunit alpha inhibitor, FIH1) in keratinocyte differentiation via Notch activation [85], miR-31 is one of the most dynamic miRNAs upregulated in psoriatic lesions, mostly in keratinocyte proliferation. Protein phosphatase 6 (PPP6C), which is inhibited by NF-κB activation [72] and large tumor suppressor kinase 2 (LATS2) [73], has been identified as a direct target of miR-31 in keratinocyte proliferation of psoriasis. The role of miR-744-3p upregulation in keratinocyte proliferation has been demonstrated via inhibition of killin (p53-regulated DNA replication inhibitor, KLLN) [77]. MiR-17-92 promotes keratinocyte proliferation via suppression of cyclin-dependent kinase inhibitor 2B (CDKN2B) [78]. The upregulation of miR-130a enhances keratinocyte proliferation by targeting serine/threonine kinase 40 (STK40) in psoriasis [79].

#### 2.2.2. Wound Healing

Wound healing is a basic biological phenomenon involving inflammation, proliferation, and remodeling in order to restore the integrity of the skin. The role of miRNAs in the proliferation phase has been suggested. Downregulated miR-96-5p promotes keratinocyte proliferation by targeting the BCL2 interacting protein 3 (BNIP3) [64]. 

The upregulation of miR-21 and miR-31 has been associated with keratinocyte proliferation phase as in psoriasis. The role of upregulated miR-21 has been identified in studies using adipose-derived stem cell exosomes containing miR-21 or miR-21 mimic-loaded nanocarriers [69,70]. Upregulation of miR-31 promotes keratinocyte proliferation during wound healing by targeting epithelial membrane protein 1 (EMP-1) [74]. Other direct targets of miR-31 include ras p21 protein activator 1 (RASA1), sprouty-related EVH1 domain containing 1 (SPRED1), SPRED2, and sprouty homolog 4 (SPRY4) [75]. Upregulated miR-17-3p stimulates keratinocyte proliferation by targeting myotilin (MYOT) via activation of the Notch1/NF-κB signaling pathway [82]. Upregulation of miR-126 enhances keratinocyte proliferation by repressing polo-like kinase 2 (PLK2) [83]. 

#### 2.2.3. Cutaneous Squamous Cell Carcinoma

cSCC is one of the most common epidermal keratinocyte-derived skin tumors. Downregulation of miR-181a in cSCC promotes keratinocyte proliferation by targeting proto-oncogene Kirsten rat sarcoma 2 viral oncogene homolog (KRAS) [29]. The increased expression of miR-181a during keratinocyte differentiation induced by high calcium or UVA irradiation [28,29] suggests that the imbalance between keratinocyte proliferation and differentiation has a considerable impact on skin barrier dysfunction. RAS is a small GTPase protein that mediates the reversible conversion between GDP-loaded inactive and GTP-loaded active forms. Human *RAS* genes encode KRAS, HRAS and NRAS isoforms. The Harvey rat sarcoma viral oncogene homolog (Hras) can be expressed upstream of miRNA instead of serving as a target of miRNA. Hras in mouse keratinocytes represses miR-203, which is the most upregulated gene during keratinocyte differentiation [37], leading to cell proliferation [86]. The role of downregulated miR-146a in SCC has been identified by activating the EGFR target [63], similar to psoriasis. 

As in psoriasis and cutaneous wound healing [70,72,74], miR-31 is upregulated in cSCC promoting cell proliferation, although the target is different: rho-related BTB domain containing 1 (RhoBTB1) [76]. Epidermal differentiation and stratification are regulated by a pool of transcription factors, including grainyhead-like 3 (GRHL3) [44]. Grhl3 represses miR-21, one of the miRNAs upregulated in cSCC as in psoriasis. Upregulation of miR-21 leads to impaired epidermal differentiation and SCC progression in Grhl3-depleted mice by targeting MutS homolog 2 (MSH2), whose sensitivity is increased by DND miRNA-mediated repression inhibitor 1 (DND1) RNA-binding protein [71].

#### 2.2.4. HPV Infection

Condyloma accuminatum caused by human papilloma virus (HPV) infection is another condition characterized by abnormal keratinocyte proliferation. Downregulation of miR-99b has been identified as a mechanism in keratinocyte proliferation via upregulation of insulin-like growth factor 1 receptor (IGF-1R) gene [65]. The downregulation of miR-203 inhibits keratinocyte differentiation and decreases involucrin expression via upregulation of ΔNp63α in epidermodysplasia verruciformis following HPV8 infection [66].

### 2.3. Structural Proteins of Corneocytes and Cornified Envelopes

Keratin-binding protein filaggrin is cross-linked into the cornified envelopes, which act as a critical structural barrier at the outermost layer of the skin epidermis. As the most insoluble components formed beneath the plasma membrane of corneocytes, the cornified envelopes are composed of various molecules, such as involucrin, envoplakin, periplakin, loricrin, elafin, S100, small proline-rich proteins, and late envelope proteins cross-linked by transglutaminases. 

Transglutaminases are a family of calcium-dependent enzymes cross-linking several structural proteins to complete the formation of cornified envelope. Among nine transglutaminases identified in human skin, transglutaminases 1, 3, and 5 are involved in cornified envelope formation [87], as shown in acral peeling skin syndrome mediated via missence mutations in transglutaminase 5 [88] and lamellar ichthyosis in transglutaminase 1 deficiency [89,90]. Elafin is highly expressed in keratinocytes of patients with psoriasis and epidermal skin tumors as well as after injury and exposure to ultraviolet (UV) radiation. Small proline-rich proteins are components of structural proteins in cornified envelopes. During the process of terminal differentiation, small proline-rich protein 1 (SPRR1) is expressed at earlier stages than SPRR2. SPRR1 is necessary to differentiate keratinocytes either by increasing extracellular calcium or 12-*O*-tetradecanoylphorbol-13-acetate. However, limited data suggest not only the association of transglutaminases 1, 3, 5, elafin, and small proline-rich proteins with miRNAs, but also that of transcription factors for each of these components with miRNAs.

The association between miRNAs and the other structural proteins with their transcription factors or epidermal barrier function was reviewed (Table 3).

#### 2.3.1. Filaggrin, Involucrin, Loricrin and their Transcription Factors

Filaggrin plays a critical role in the structural and mechanical integrity of stratum corneum as shown in loss-of-function mutations of human filaggrin gene in defective barrier function of atopic dermatitis [98,99]. Although filaggrin is not one of the cornified envelope proteins, it has been frequently investigated with involucrin and loricrin, major components of cornified envelope proteins.

Instead of the association of these components with miRNAs as direct targets or regulators, the altered expression of filaggrin, involucrin, and loricrin is accompanied by changes in other differentiation markers and barrier proteins as described in keratinocyte differentiation associated with upregulation of miR-203 and miR-184 [20,23,32]. Downregulation of miR-339-5p increases levels of distal-less homeobox5 (DLX5), leading to involucrin upregulation via activation of the Wnt/β-catenin signaling pathway [91]. Dicer is involved in the regulation of skin barrier function [100]. Based on Dicer acting as a target of miR-107 [101], the role of miR-107 in epidermal barrier has been investigated in the trial drug delivery of anti-miR-107 into skin following burn injury. The downregulated miR107 increased the expression of barrier proteins including filaggrin and loricrin via Dicer upregulation [92].

The bulk of existing miRNA studies in allergic diseases including atopic dermatitis focus mainly on dysregulation of immune system [10,11]. Based on the action mechanism of miR-143 identified in nasal epithelial cells of allergic rhinitis patients [102], the role of miR-143 in filaggrin, loricrin and involucrin has been identified [10]. The downregulation of miR-143 in atopic dermatitis stimulates the activity of IL-13 secreted by activated Th2 cells by targeting interleukin 13 receptor subunit α1 (IL-13Rα1), resulting in reduced expression of filaggrin and components of cornified envelope proteins such as involucrin and loricrin [12,93,94].

The expression of filaggrin and a subset of cornified envelope genes including involucrin and loricrin is regulated by various transcription factors including MafB, signal transducer and activator of transcription 3 (STAT3), GATA transcription factor 3 (GATA3), OVO-like protein 1 (OVOL1) or 6-formylindolol[3,2-b]-carbazole (FICZ) via aryl hydrocarbon receptor (AHR), AP-1, putative zinc transporter 10 (ZIP10), histone acetyltransferases (HATs), early growth response 3 (EGR3), forkhead box O1 (FOXO1), and transcription factor 7-like 1 (TCF7L1)/ transcription factor 4 (TCF4). However, limited data suggest the association of transcription factors involved in keratinocyte differentiation with miRNAs. They include upregulation of miR-203 by AP-1 activation [37], which was previously mentioned in connection with calcium. The CCAAT/enhancer binding protein α (C/EBPα) is a transcription factor regulating differentiation via directly binding and transcription of miR-203. The suppression of C/EBPα/miR-203 pathway by HPV8 E6 protein leads to the downregulation of involucrin expression via activation of ΔNp63α, a target of miR-203 [66].

#### 2.3.2. S100 and its Transcription Factors

Gene families of S100A and S100-fused genes are the main components of epidermal differentiation complex. S100A7 acts as a transglutaminase substrate/cornified envelope precursor. S100A7 is also known as psoriasin, which is increased in epidermal hyperproliferative disorders [103]. Downregulation of miR-6731-5p activates S100A7, which is involved in IL-22-stimulated keratinocyte proliferation [96].

As for the regulation of S100 gene families, studies reported the upregulation of IL-17-induced psoriasis-associated genes including S100A7 via IκBζ, downregulation of S100A7 by thymic stromal lymphopoietin (TSLP), binding of CCAAT/enhancer binding protein β (C/EBPβ) to the promoter region of S100A8, and the induction of S100SA2 protein, an important regulator of keratinocyte differentiation, by p53 tumor suppressor protein. However, the association between these regulatory factors in S100 gene families and miRNAs has yet to be demonstrated.

#### 2.3.3. Others

Suppression of EGFR signaling facilitates terminal differentiation of keratinocytes. In addition, EGFR is a well-known target of miR-146a and is repressed via upregulation of miR-146a, thereby inhibiting keratinocyte proliferation [63]. As described in connection with psoriasis, the role of miR-146a/EGFR axis in keratinocyte proliferation has been reported [62]. The exacerbation of EGFR degradation by reduced miR-30a-3p has been shown to result in abnormal keratinocyte differentiation in patients with familial acne inversa [97].

Despite the critical role of filaggrin mutation in barrier dysfunction of patients with atopic dermatitis, the association of miRNAs with barrier abnormalities in this disease, independent of filaggrin, has been also suggested. The downregulation of let-7a-5p may be related to barrier abnormalities by targeting ribonucleotide reductase regulatory subunit M2 (RRM2) and C-C motif chemokine receptor 7 (CCR7) [95], which are mainly involved in cancer cell proliferation. Downregulated miR-26a-5p is also involved in barrier function of patients with atopic dermatitis [95] by targeting hyaluronan synthase 3 (HAS3), DEP domain-containing 1B (DEPDC1B), DEPDC1, nicotinamide phosphoribosyltransferase (NAMPT), DENN domain-containing 1B (DENND1B), and a disintegrin and metalloproteinase domain 19 (ADAM19), which mediate cell differentiation, cell proliferation, and anti-apoptosis. HAS catalyzes the synthesis of hyaluronan glycosaminoglycan, whose turnover in extracellular space of the vital epidermis is enhanced during epidermal differentiation [104]. Although HAS3 is one of the targets identified in downregulated miR-26a-5p [95], it is a direct target of upregulated miR-10a-5p in atopic dermatitis [12]. Upregulated miR-29b increases interferon-γ-induced keratinocyte apoptosis by repressing Bcl-2-like 2 (BCL2L2) [12].

## 3. Role of miRNAs in Cell-Cell Adhesion

The cell–cell adhesion of corneocytes depends on corneodesmosomes, which are modified desmosomes formed via differentiation of keratinocytes from granular cell layers into cornified cell layers. Corneodesmosomes are involved in desquamation, which is the gradual invisible shedding of corneocytes. Desquamation is determined by de novo synthesis and degradation of corneodesmosomal proteins. The key role of desquamation in epidermal barrier integrity and homeostasis has been attributed to accelerated desquamation either via reduced corneodesmosome synthesis or increased corneodesmosome degradation linked to skin disorders with barrier dysfunction [105]. The association between miRNAs and structural proteins of corneodesmosome or proteases and inhibitors involved in corneodesmosome degradation has been reviewed (Table 4).

### 3.1. Synthesis of Corneodesmosomes

The components of corneodesmosomes include desmoglein 1, desmocollin 1, and corneodesmosin. Desmoglein 1, concentrated in the uppermost cell layers of epidermal keratinocytes, is a member of the cadherin family of calcium-dependent cell adhesion molecules, with a significant impact on keratinocyte differentiation. Long noncoding RNAs (lncRNAs) regulate miRNAs. The regulatory role of H19 and miR-130b-3p interaction in desmoglein 1 expression has been investigated based on upregulation of H19 lncRNA but downregulation of miR-130b-3p in keratinocytes treated with calcium [106]. The finding suggests that H19 inhibits miR-130b-3p, and thereby increases the expression of desmoglein 1, a direct target of miR-130b-3p. 

### 3.2. Degradation of Corneodesmosomes

The degradation of corneodesmosomal proteins is controlled by proteases and a variety of inhibitors. Kallikrein-related peptidases (KLKs) represent a family of serine proteases. KLKs are secreted as precursors and activated upon proteolytic cleavage. Although 15 different serine proteases belonging to the KLK family have been detected in normal human skin [108], KLKs show tissue-specific expression. KLKs 5, 7, 11, and 14 are highly expressed in human skin and KLKs 5, 7, and 14 play a role in the degradation of corneodesmosomal proteins [109]. Epigenetic changesin miRNAs have focused the attention on the control of KLK expression. Cornification has been delayed in mice with protease-activated receptor type 2 (PAR2) ablation [110], indicating the role of PAR2 in epidermal barrier integrity. Although PAR2 is activated via proteolytic cleavage by KLK5, few studies have investigated the association between miRNAs and the KLKs involved in corneodesmosome protein degradation.

The activity of KLK serine proteases is controlled by proteinaceous endogenous KLK inhibitors, Kazal-type inhibitors, and macroglobulins. KLKs 5, 7, and 14 are inhibited by serpins, such as α1-antitrypsin, α1-chymotrypsin, antithrombin III, α2-antiplasmin, heparin-dependent proteinase C and kallistatin, and lymphoepithelial-Kazal-type 5 inhibitor (LEKTI), or α2-macroglobulin. In addition to serine proteases, other proteases such as plasmin, plasma kallikrein, factor Xa, or plasminogen activator (tissue-type or urokinase-type) are involved in the activation of KLKs. The association between miRNAs and these inhibitors in epidermal barrier has yet to be reported. 

### 3.3. Others

Desmosomes are multi-protein adhesive complexes connecting adjacent keratinocytes. Desmosomal adhesion is also important to ensure the integrity and protective barrier function of the epidermis. Desmosomal cadherins are composed of desmocollins and desmogleins, which include several isoforms. Isoform 3 of desmocollin and desmoglein is expressed strongest in the basal proliferative layer, whereas isoform 1 expression is predominant in the upper layers of epidermis. Misdirection of these protein isoforms affects keratinocyte differentiation, resulting in altered β-catenin stability [111]. A link between desmosomal cadherin and the stability of β-catenin, which is negatively regulated by miR-214 as a direct target [107], suggests the role of miR-214 in epidermal barrier function.

## 4. Role of miRNAs in Skin Lipids

Skin lipids, which constitute the extracellular matrix of the SC, are composed of cholesterol, free fatty acids, and ceramides. These lipids are stacked to form densely packed lipid layers, lipid lamellae depending on the composition of the lipids. Precursor lipids are synthesized in keratinocytes, although lipids derived from sebaceous glands and extracutaneous sources also contribute to the epidermal lipid pool [112,113]. Ceramides form the structural backbone of sphingolipids, contributing to the structural diversity particularly in epidermis. Changes in ceramide levels, composition and chain length are the most distinctive hallmark of atopic dermatitis [114,115]. Sphingomyelin generated by sphingomyelin synthase (SGMS) and glucosylceramides synthesized by UDP-glucose ceramide glucosyltransferase (UGCG) are precursors of ceramide. The pathogenesis of various ichthyoses and ichthyosis syndromes is mediated via abnormal synthesis of ceramides, particularly ultra-long-chain acylceramide. NIPA-like domain containing 4 (NIPAL4), CYP4F22, arachidonate 12-lipooxygenase, 12R type (ALOX12B) and arachidonate lipoxygenase 3 (ALOXE3) catalyze the synthesis of acylceramides and their precursor ω-hydroxyceramides. Fatty acid elongase 4 (ELOVL4), fatty acid transporter protein 4 (FATP4), cytochrome P450 4F22 (CYP4F22), patatin-like phospholipase domain-containing 1 (PNPLA1), and α/β hydrolase domain containing protein 5 (ABHD5) are required for the biosynthesis of ω-О-acylceramides, whereas 12R-lipooxygenase (12R-LOX), epidermis-type lipooxygenase 3 (eLOX3), and short-chain dehydrogenase/reductase family 9C member 7 (SDR9C7) catalyze the transformation of ω-О-acylceramides to covalently bound ceramides that form the corneocyte lipid envelope [116]. NADP-dependent steroid dehydrogenase-like (NSDHL) for the synthesis of cholesterol, another major component of skin lipid, is also related to skin barrier function. The ATP-binding cassette transporter A12 (ABCA12), vacuolar protein sorting 33 homolog B (VPS33B) and VPS33B interacting protein, apical-basolateral polarity regulator, spe-39 homolog (VIPAS39), cystatin A (*CSTA*)*,* Rab11a GTPase, and fatty acid transport protein 4 (FATP4) are involved in lamellar body formation and secretion.

Peroxisome proliferator-activated receptor (PPAR) isoforms (alpha, beta/delta, and gamma) and liver X receptor (LXR) isoforms are expressed in the epidermis. Activation of these receptors stimulates epidermal lipid synthesis, lamellar body formation and secretion, and activation of enzymes involved in extracellular processing of lipids in the stratum corneum [117]. However, few studies investigated the association between miRNAs and skin lipids, which include not only the main components but also precursors and enzymes, the components for lamellar body formation and secretion, and related receptors (Table 5).

### 4.1. Skin Lipids

The association between miRNA and ALOX12B enzymes involved in extracellular processing of lipids has been reported. MiR-185-5p, which is upregulated by p-ΔNp63, targets ALOX12B 3’-UTR with reduced activity [118]. In the context of calcium-induced keratinocyte differentiation, oleic acid, an unsaturated free fatty acid constituent of sebum, has been shown to accelerate keratinocyte differentiation via upregulation of miR-203 [20]. Linoleic acid and ciglitazone also increased sebaceous lipogenesis via upregulation of miR-203 and miR-574-3p, which are also implicated in keratinocyte differentiation [119].

### 4.2. Epidermal Receptors Binding Skin Lipids

The outcome of nuclear hormone receptor PPARβ/δ activity relies on transcriptional activation of target genes. Excessive exposure to UV, a major causative factor for skin cancer, upregulated miR-21-3p in a PPARβ/δ-dependent manner in the epidermis of Ppard^+/+^ mice and in human keratinocytes. In the absence of a PPAR-binding site in the promoter of miR-21-3p and abolished PPARβ/δ-dependent miR-21-3p activation via TGFβ receptor inhibition, SMAD7, an antagonist of TGFβ1 signaling, has been identified as a direct target of miR-21-3p [120]. The dominant role of PPARβ/δ/ TGFβ1/miR-21-3p cascade is related to the regulation of immune and inflammatory responses. However, microarray profiling of human keratinoc ytes overexpressing miR-21-3p also demonstrated the downregulated expression of caspase-14, a member of the caspase family associated with keratinocyte differentiation and barrier formation, indicating a role in UV-mediated epidermal barrier disruption [120].

## 5. Conclusions

Formation and maintenance of skin barrier integrity under normal and pathological conditions is influenced by genetic and environmental factors involved in epidermal cell differentiation and proliferation, cell–cell adhesion, and skin lipids. Gene expression is regulated by miRNAs at the post-transcriptional level by triggering RNA interference. The critical role of miRNAs in the epidermal barrier expands the use of miRNAs for unmet clinical needs in the future. The current efforts are driven by the need to combat various diseases using miRNA-based technologies. However, miRNA targeting is associated with a significant risk of undesirable effects. The risk can be overcome by maximizing delivery to the target organ and minimizing off-target effects. Topical delivery is one of the easiest strategies for targeted delivery and skin disorders can be greatly ameliorated using targeted miRNA therapy [55,121,122]. Thus, this review highlights the association between miRNAs and factors involved in epidermal barrier integrity, including regulatory factors and structural constituents associated with keratinocyte differentiation and proliferation, desquamation of corneodesmosomes, and skin lipids.

Extracellular calcium, p63 and Notch signaling can upregulate or downregulate miRNAs in keratinocyte differentiation and proliferation. The miRNAs can also directly target p63 and Notch. The balance between keratinocyte proliferation and differentiation in the intact skin barrier is negative in skin disorders with aberrant keratinocyte proliferation. In fact, similar miRNAs may be involved in both keratinocyte differentiation and proliferation depending on upregulation or downregulation of the corresponding miRNAs. The miR-203 is the first and the most upregulated miRNA underlying epidermal differentiation. Multiple proteins, such as SNAI2 and ΔNp63 [18], p63 [19,20], and p63, Skp2 and Msi2 in mouse skin [21], play a role in keratinocyte differentiation induced by upregulation of miR-203 as direct targets. However, the downregulation of miR-203 contributes to keratinocyte proliferation in epidermodysplasia verruciformis by targeting ΔNp63, either (Figure 1). The effect of upregulation and downregulation of similar miRNA on keratinocyte differentiation and proliferation has also been demonstrated with miR-125b, miR-146a, and miR-181a (Figure 2). 

Several miRNAs have been identified in skin diseases associated with keratinocyte proliferation, such as psoriasis, wound healing, skin cancer, and warts. All these diseases share specific miRNAs including miR-21 and miR-31 related to epidermal barrier function. The upregulation of miR-21 and miR-31 promotes keratinocyte proliferation in psoriasis, wound healing and cSCC, although different therapeutic targets have been identified (Figure 3A,B).

Mutations in filaggrin and structural proteins of cornified envelopes have also contributed to the pathogenesis of representative skin diseases associated with primary barrier dysfunction. However, miRNA studies involving allergic diseases including atopic dermatitis focus mainly on the dysregulation of immune system [10,11]. Limited data suggest the association of structural proteins in corneocytes, cornified envelopes, and their transcription factors with miRNAs. Instead of the association of these components with miRNAs as direct targets or regulators, the change in expression of each component including filaggrin, involucrin, and loricrin is mostly accompanied by changes in other differentiation markers and barrier proteins. Although atopic dermatitis is a representative skin disorder with barrier abnormalities, particularly related to filaggrin, the relevant miRNAs have been implicated in cell differentiation, proliferation, and anti-apoptosis in connection with barrier abnormalities. 

Limited evidence is also available to support the association between miRNAs and cell–cell adhesion or skin lipids, despite extensive review of the structural proteins in corneodesmosome, proteases and inhibitors involved in corneodesmosome degradation, lipid composition of skin, enzymes involved in lipid synthesis and secretion, and receptors involved in epidermal lipid metabolism. Although abnormal cell–cell adhesion associated with accelerated desquamation and defective skin lipids play a role in the development of ichthyoses [105], the role of miRNAs in ichthyoses has yet to be reported.

## Figures and Tables

**Figure 1 ijms-21-05781-f001:**
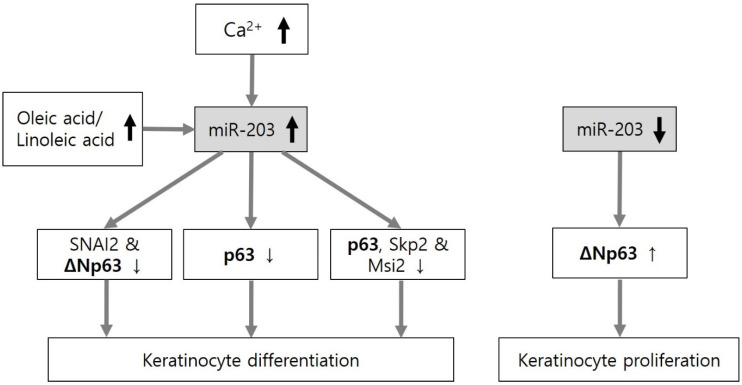
Role of miR-203 in keratinocyte differentiation and proliferation. As the first and most upregulated miRNA under high calcium concentrations or exposure to unsaturated fatty acids (oleic acid or linoleic acid), miR-203 can control both keratinocyte differentiation and proliferation via downregulation and upregulation of ΔNp63, respectively.

**Figure 2 ijms-21-05781-f002:**
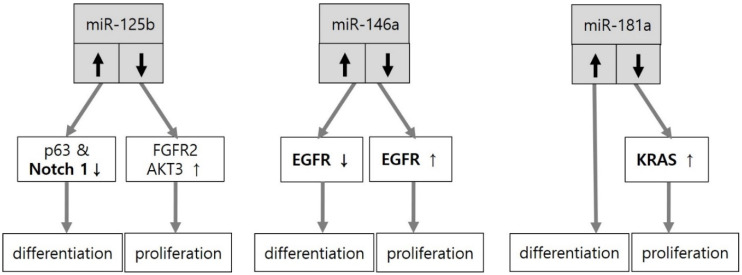
miRNAs involved in both keratinocyte differentiation and proliferation depending on their expression. Upregulation of miR-125b stimulates keratinocyte differentiation via suppression of both p63 and Notch1 expression, whereas its downregulation promotes keratinocyte proliferation by targeting fibroblast growth factor receptor 2 (FGFR2) or AKT serine/threonine kinase 3 (AKT3). Downregulation of miR-146a facilitates keratinocyte proliferation by activating epidermal growth factor receptor (EGFR), whereas its upregulation promotes terminal differentiation of keratinocytes by suppressing EGFR. Downregulation of miR-181a promotes keratinocyte proliferation by targeting Kirsten rat sarcoma 2 viral oncogene homolog (KRAS). However, the increased expression of miR-181a by high calcium or UVA irradiation is associated with keratinocyte differentiation.

**Figure 3 ijms-21-05781-f003:**
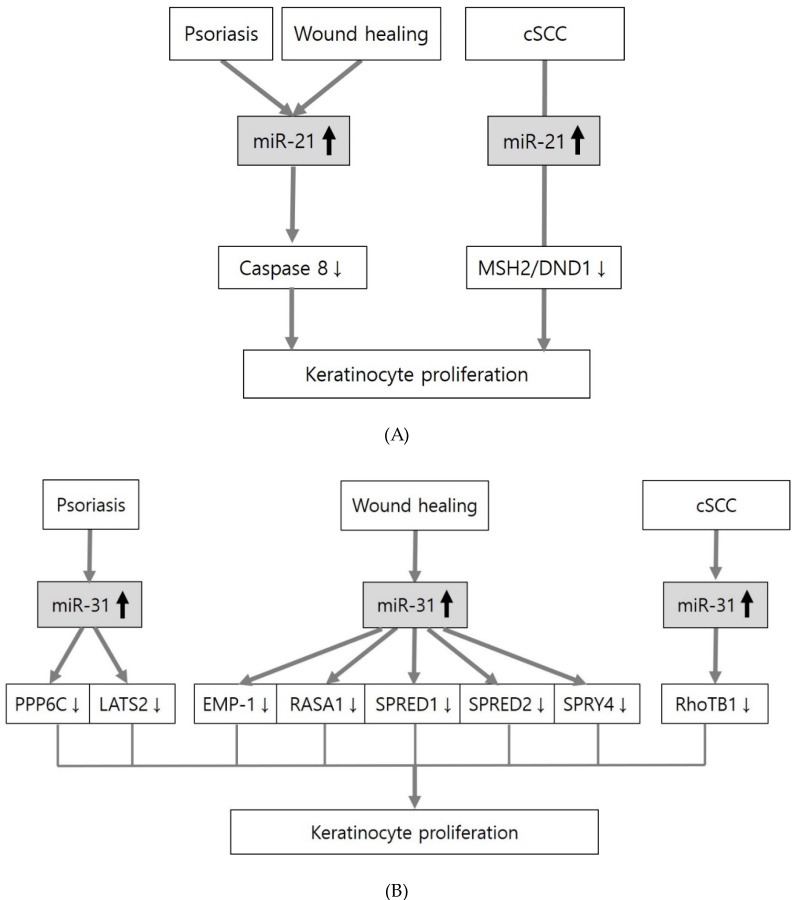
Association of miR-21 and miR-31 in skin diseases with distinct keratinocyte proliferation. A distinct proliferation of keratinocytes occurs in psoriasis, the proliferation phase of wound healing, and cutaneous squamous cell carcinoma (cSSC). Upregulation of miR-21 (**A**) and miR-31 (**B**) promotes keratinocyte proliferation in several miRNAs identified in these skin diseases associated with keratinocyte proliferation, although the identified targets vary.

**Table 1 ijms-21-05781-t001:** miRNAs related to regulatory factors underlying keratinocyte differentiation and proliferation.

Regulatory Factor	miRNA	Target Molecule	Action Mechanism of miRNA	Reference
Calcium ↑	miR-203 ↑	SNAI2 and ΔNp63	Activation of the PKC and AP-1 pathway	[18]
		p63	Upregulation of JNK by galectin-7	[19]
			Increased keratinocyte differentiation with involucrin expression by oleic acid	[20]
		p63, Skp2 and Msi2	Promotion of cell cycle exit in mouse skin	[21]
	miR-574 ↑	p63	As direct targets of iASPP	[22]
	miR-720 ↑			
	miR-24 ↑	PAK4	Control of actin cable formation	[23]
	miR-23b-3p ↑	TGIF1	Interference in TGF-β/SMAD signaling	[24]
	miR-378b ↑	NKX3.1		[25]
	miR-30a ↑	LOX, IDH1, AVEN	Barrier function defects in aged epidermis	[26]
	miR-184 ↑		Upregulation of cyclin E and p21 cyclin-dependent kinase inhibitor in a SOCE-dependent manner	[27]
	miR-181a ↑		cell differentiation under high calcium or UVA exposure	[28,29]
	miR-125b ↑	p63	cell differentiation and proliferation in Hailey-Hailey disease	[30]
		Notch		
p63 ↑	miR-944 ↑		Upregulation of K1 and K10 by ERK inhibition and p53 upregulation	[31]
p63 ↓	miR-17/miR-20b /miR-106a ↓	p21, RB, and JNK2	Upregulation of K1 and K10	[32]
Notch ↑	miR-184 ↑	K15 and FIH1	Enhancing the Notch pathway	[33]
miRNAs targeted by p63	miR-34a ↑	SIRT6	miR-34a and miR-34c as direct targets of p63	[34,35]
	miR-34a ↑	KLK4	Induction of a senescent phenotype in keratinocytes	[36]

↑: upregulation (increase), ↓: downregulation (decrease).

**Table 2 ijms-21-05781-t002:** miRNAs related to keratinocyte proliferation.

miRNA		Target Molecule	Related Skin Diseases	Reference
Change	Name			
↓	miR-125b	FGFR2	Psoriasis	[46]
		AKT3		[47]
	miR-181b-5p	AKT3, TRL4		[47,48]
	miR-520a	AKT		[49]
	miR-320b	AKT3		[50]
	miR-194	GRHL2		[51]
	miR-217			[52]
	miR-138	hTERT		[53]
	miR-150	HIF-1α, VEGFA		[54]
	miR-145-5p	MLK3		[55]
	miR-20a-3p	SFMBT1		[56]
	miR-876-5p	ANG-1		[57]
	miR-99a	FZD5/FZD8		[58]
	miR-187	CD276		[59]
	miR-548a-3p	PPP3R1		[60]
	miR-330	CTNNB1		[61]
	miR-146a	EGFR	Psoriasis, cSCC	[62,63]
	miR-96-5p	BNP3	Wound healing	[64]
	miR-181a	KRAS	cSCC	[29]
	miR-99b	IGF-1R	Condyloma acuminatum	[65]
	miR-203	ΔNp63	Epidermodysplasia verruciformis	[66]
↑	miR-21	Caspase 8	Psoriasis	[67,68]
			Wound healing	[69,70]
		MSH2	cSCC	[71]
	miR-31	PPP6C	Psoriasis	[72]
		LATS2		[73]
		EMP-1	Wound healing	[74]
		RASA1		[75]
		SPRED1		
		SPRED2		
		SPRY4		
		RhoTB1	cSCC	[76]
	miR-744-3p	KLLN	Psoriasis	[77]
	miR-17-92	CDKN2B		[78]
	miR-130a	STK40		[79]
	miR-122-5p	SPRY2		[80]
	miR-223	PTEN		[81]
	miR-17-3p	MYOT	Wound healing	[82]
	miR-126	PLK2		[83]

↑: upregulation, ↓: downregulation.

**Table 3 ijms-21-05781-t003:** miRNAs associated with structural proteins of corneocytes and cornified envelopes or epidermal barrier function.

miRNA	Target Molecule	Related Skin Disease	Action Mechanism of miRNA	Reference
miR-339-5p ↓	DLX5		Increased involucrin expression through Wnt/β-catenin signaling pathway activation	[91]
miR-107 ↓	Dicer		Increased filaggrin and loricrin expression	[92]
miR-203 (C/EBPα/miR-203 pathway) ↓	ΔNp63α	HPV8 infection	Downregulation of involucrin	[66]
miR-143 ↓	IL-13Rα1	Atopic dermatitis	Reduced filaggrin/ involucrin /loricrin expression through Th2-derived IL-13 activity stimulation	[12,93,94]
Let-7a-5p ↓	RRM2, CCR7		Barrier abnormalities	[95]
miR-26a-5p ↓	HAS3, DEPDC1B, DEPDC1, NAMPT, DENND1B, ADAM19			[95]
miR-10a-5p ↑	HAS3			[12]
miR-29b ↑	BCL2L2		Barrier abnormalities with Increased IFN-γ-induced keratinocyte apoptosis	[12]
miR-6731-5p ↓	S100A7		IL-22-stimulated keratinocyte proliferation	[96]
miR-146a ↑	EGFR		Terminal differentiation and proliferation inhibition in keratinocytes	[62]
miR-30a-3p ↓		Familial acne inversa	Abnormal keratinocyte differentiation by accelerated EGFR degradation	[97]

↑: upregulation, ↓: downregulation.

**Table 4 ijms-21-05781-t004:** miRNAs associated with structural proteins of corneodesmosomes or proteases and inhibitors involved in corneodesmosome degradation.

miRNA	Target Molecule	Action Mechanism of miRNA	Reference
miR-130b-3p ↓	Desmoglein 1 ↑	miR-130b-3p is inhibited by upregulated H19 in keratinocytes treated with calcium	[106]
miR-214 ↑	β-catenin ↓	β-catenin is linked to desmosomal cadherins, resulting in epidermal barrier dysfunction.	[107]

↑: upregulation, ↓: downregulation.

**Table 5 ijms-21-05781-t005:** miRNAs associated with skin lipids.

miRNA	Target Molecule	Related Skin Condition	Action Mechanism of miRNA	Reference
miR-185-5p ↑	ALOX12B		Upregulated by p-ΔNp63, resulting in reduced ALOX12B activity.	[117]
miR-203 ↑			Upregulated by oleic acid, promoting keratinocyte differentiation	[20]
miR-203 ↑			Upregulated by linoleic acid, stimulating keratinocyte differentiation	[118]
miR-574-3p ↑				
miR-21-3p ↑	SMAD7	UV exposure	Upregulated in a PPARβ/δ-dependent manner Besides regulatory role in immune and inflammatory responses, miR-213p can downregulate caspase-14 expression. (--Arrangement was changed as a center alignment array)	[119]

↑: upregulation.

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
