# Peer review of "The Role of MicroRNAs in Epidermal Barrier"

_ijms, 2020, doi:10.3390/ijms21165781_

Round 1

Reviewer 1 Report

This is an exhaustively research review of the role of miRNAs in the control of skin cell function, primarily keratinocyte proliferation and differentiation.  The writing is clear and very concise. 

There is so much data in this manuscript that relates to the role of miRNAs in not only barrier properties, but their role in skin cancer, psoriasis, wound healing, etc. that the title (although it's fine)  does not accurately reflect all the various information provided.

The authors might want to consider citing the figures earlier in the text rather than in the Conclusion section. Having a visual to look at could help the reader better understand, for example, the role of miRNA-203 in regulating keratinocyte growth vs differentiation. 

In the legend to figure 1, please check to make sure that the wording "...via downregulation and downregulation of ANp63, respectively" is correct. It appears as though "p63" may have been left off accidentally.

One area that is not covered in this review is how miRNAs themselves are regulated. How is their synthesis regulated and by what pathways? However, given the extensive amount of data already in the paper, I would not suggest trying to include this additional material. 

Author Response

The manuscript titled “The Role of MicroRNAs in Epidermal Barrier” has been revised according to reviewers’ comments as followings.

As for the recommendation of Reviewer 1;

There is so much data in this manuscript that relates to the role of miRNAs in not only barrier properties, but their role in skin cancer, psoriasis, wound healing, etc. that the title (although it's fine) does not accurately reflect all the various information provided.

Response) In order to exclusively include the miRNAs related to epidermal barrier in skin cancers, psoriasis, wound healing…, it was considered whether the corresponding miRNAs have been related to keratinocyte differentiation in vivo and in vitro, whether the function of their targets have been related to keratinocyte differentiation or proliferation, and so on. I tried to exclude miRNAs mostly related to inflammation, immune reaction, remodeling, and so on. These rules were hold in case of atopic dermatitis, either.

The authors might want to consider citing the figures earlier in the text rather than in the Conclusion section.

Having a visual to look at could help the reader better understand, for example, the role of miRNA-203 in regulating keratinocyte growth vs differentiation.

Response) I absolutely agree with your opinion. However, it was difficult to find out such place in the main text.

In the legend to figure 1, please check to make sure that the wording "...via downregulation and downregulation of ANp63, respectively" is correct. It appears as though "p63" may have been left off accidentally.

Response) As you pointed out, the legend <via downregulation and downregulation of ΔNp63, respectively> was wrong. It should have been <via downregulation and upregulation of ΔNp63, respectively>.

One area that is not covered in this review is how miRNAs themselves are regulated. How is their synthesis regulated and by what pathways? However, given the extensive amount of data already in the paper, I would not suggest trying to include this additional material.

Response) I appreciate your advice.

All the revised portions were highlighted in yellow in the manuscript made formatting change by the editorial office.

Sincerely Yours,

Ai-Young Lee, M.D., Ph.D.

Professor

Department of Dermatology, College of Medicine, Dongguk University Ilsan Hospital

814, Siksa-dong, Ilsandong-gu, Goyang-si, Gyeonggi-do, 410-773, Republic of Korea

Tel: +82-31-961-7250

Fax: +82-31-961-7695

E-mail: lay5604@naver.com

Reviewer 2 Report

This manuscript by Lee provides an up-to-date review on the role of miRNAs in the epidermal barrier. Overall, this manuscript is well-written and provides an informative review of the existing literature on an important topic, and it may be used to inform future research in this area. I have included some minor suggestions to improve the clarity of the manuscript.

Lines 58-61
References are required for the sentence beginning by "Several studies have …"

Lines 68-69
structural proteins of corneocytes and cornified envelopes?

Line 190
A typo needs to be fixed: condyloma acuminatum

Lines 446-451
ELOVL4 , FATP4 , CYP4F22 , PNPLA1, and ABHD5 are required for the biosynthesis of ω-О-acylceramides, whereas 12R-LOX, eLOX3, and SDR9C7 catalyze the transformation of ω-О-acylceramides to covalently bound ceramides that form the corneocyte lipid envelope (Takeichi et al. J Clin Invest, 2020). In addition, PNPLA1 stands for Patatin-like phospholipase domain-containing 1 or Patatin-like phospholipase domain-containing protein 1.

Figure 3
What is the difference between Figure 3a and 3b?
In the right side, sSCC, miR-21, and MSH2/DND1 should be connected by the arrows with heads.

Figure 4 is not cited in the text.

Author Response

The manuscript titled “The Role of MicroRNAs in Epidermal Barrier” has been revised according to reviewers’ comments as followings.

As for the recommendation of Reviewer 2;

Lines 58-61
References are required for the sentence beginning by "Several studies have …"

Response) Too many references should be listed as shown in psoriasis, wound healing, SCC…, <several> is changed as <more>.

Lines 68-69
structural proteins of corneocytes and cornified envelopes?

Response) As you point out, <of> should have been marked <and>. Thank you for your careful review.

Line 190
A typo needs to be fixed: condyloma acuminatum

Response) <condyloma accuminatum> at line 190 and Table 2 is corrected as <condyloma acuminatum>.

Lines 446-451
ELOVL4 , FATP4 , CYP4F22 , PNPLA1, and ABHD5 are required for the biosynthesis of ω-О-acylceramides, whereas 12R-LOX, eLOX3, and SDR9C7 catalyze the transformation of ω-О-acylceramides to covalently bound ceramides that form the corneocyte lipid envelope (Takeichi et al. J Clin Invest, 2020).

Response) The sentences you pointed out are changed as you described with the reference you gave me as [116]. The following numbers (original reference numbers from 116 to 121) were raised their original numbers by one.

In addition, PNPLA1 stands for Patatin-like phospholipase domain-containing 1 or Patatin-like phospholipase domain-containing protein 1.

Response) The full name of PNPLA1 is changed as <patatin-like phospholipase domain-containing 1>.

Figure 3
What is the difference between Figure 3a and 3b?
In the right side, sSCC, miR-21, and MSH2/DND1 should be connected by the arrows with heads.

Response) The portion you pointed out is revised as arrows with heads. There was no Figure 4. Instead, the schematic diagram concerning miR-21 was Figure 3a, and that concerning miR-31 was Figure 3b.

Figure 4 is not cited in the text. 

Response) As explained above, there was no Figure 4.

All the revised portions were highlighted in yellow in the manuscript made formatting change by the editorial office.

Sincerely Yours,

Ai-Young Lee, M.D., Ph.D.

Professor

Department of Dermatology, College of Medicine, Dongguk University Ilsan Hospital

814, Siksa-dong, Ilsandong-gu, Goyang-si, Gyeonggi-do, 410-773, Republic of Korea

Tel: +82-31-961-7250

Fax: +82-31-961-7695

E-mail: lay5604@naver.com
